# Multifunctional Biomimetic Composite Coating with Antireflection, Self-Cleaning and Mechanical Stability

**DOI:** 10.3390/nano13121855

**Published:** 2023-06-13

**Authors:** Zhibin Jiao, Ze Wang, Zhaozhi Wang, Zhiwu Han

**Affiliations:** 1Key Laboratory of Bionic Engineering, Ministry of Education, Jilin University, Changchun 130022, China; jiaozhibin@sut.edu.cn; 2School of Mechanical Engineering, Shenyang University of Technology, Shenyang 110870, China; zhaozhi_wang@sut.edu.cn; 3School of Mechanical and Aerospace Engineering, Jilin University, Changchun 130022, China

**Keywords:** biomimetic coating, antireflective, self-cleaning, abrasion resistance, micro-/nano-structure

## Abstract

Antireflective and self-cleaning coatings have attracted increasing attention in the last few years due to their promising and wider applications such as stealth, display devices, sensing, and other fields. However, existing antireflective and self-cleaning functional material are facing problems such as difficult performance optimization, poor mechanical stability, and poor environmental adaptability. Limitations in design strategies have severely restricted coatings’ further development and application. Fabrication of high-performance antireflection and self-cleaning coatings with satisfactory mechanical stability remain a key challenge. Inspired by the self-cleaning performance of nano-/micro-composite structure on natural lotus leaves, SiO_2_/PDMS/matte polyurethane biomimetic composite coating (BCC) was prepared by nano-polymerization spraying technology. The BCC reduced the average reflectivity of the aluminum alloy substrate surface from 60% to 10%, and the water contact angle (CA) was 156.32 ± 0.58°, illustrating the antireflective and self-cleaning performance of the surface was significantly improved. At the same time, the coating was able to withstand 44 abrasion tests, 230 tape stripping tests, and 210 scraping tests. After the test, the coating still showed satisfactory antireflective and self-cleaning properties, indicating its remarkable mechanical stability. In addition, the coating also displayed excellent acid resistance, which has important value in aerospace, optoelectronics, industrial anti-corrosion, etc.

## 1. Introduction

Aluminum alloys are widely used in the military, aerospace and transportation sectors due to their light weight, excellent thermal conductivity, high ductility and good mechanical properties. However, limited by the high reflectivity and accumulated pollutants of the aluminum alloys surface, the function of the aluminum alloy device is seriously affected. The traditional method to solve these problems have relied mainly on complex cleaning processes [1,2]. This method is time-consuming, expensive, and not thoroughly cleaned. Moreover, it increases the waste of water resources, and might damage the original material seriously.

In contrast, the self-cleaning coating provides a good solution. Ghosh et al. prepared vertically arranged tree-like nanostructures with graded roughness on polished silicon substrates, exhibiting excellent superhydrophobicity and low adhesion [3]. Wang et al. prepared ZnS superhydrophobic coating on the surface of zinc substrate by economic and environmental-friendly solvothermal method and chemical modification method [4]. Xia et al. provided a SiO_2_-TiO_2_-PDMS composite coating to demonstrates the potential application of coatings to protect architectures from detrimental atmospheric effects via a self-cleaning approach [5]. These functional coatings with antireflective and self-cleaning properties show the advantages such as high efficiency, low cost, and large-area preparation, showing significant potential for applications in stealth, optoelectronics, sensing, industrial anti-corrosion, and anti-fouling [6,7]. However, it also faces some problems, such as unclear coating design principles, single function, poor adaptability, and inefficiency. Therefore, innovative design and manufacture of high-performance antireflective and self-cleaning coatings has become a research hotspot.

Creatures in nature always provide researchers with efficient solutions. Typical biological functional surfaces (such as lotus leaves, moth eyes, cicada wings, etc.) have evolved unique micro-/nano-scale structures with outstanding advantages on antireflective and self-cleaning performance [8,9,10,11,12,13,14,15]. There has been a great deal of interest in solving the problem of wettability and optical properties of material surfaces through bionic concepts and bionic techniques. Since the structures play a non-negligible role in the function of the interfaces, the preparation of coatings with biomimetic structures is a good solution to improve the performance of the interface. However, large-area preparation of biomimetic coatings with excellent antireflective and superhydrophobic self-cleaning properties remains a challenge. Currently, the chemical/physical vapor deposition method [16,17,18], electrochemical machining method [19,20,21], and the femtosecond laser method [22,23] are mainly used to fabricate functional coatings on metal substrate. In particular, large-scale application of these techniques is difficult due to the complex preparation process, high cost, and single surface function. By contrast, the spray deposition process is a simple and efficient fabrication technique [24,25], and it mainly achieves specific functions by introducing some spherical inorganic nanoparticles (such as SiO_2_, TiO_2_) into the solution to construct micro-/nano-rough structures [26,27,28].

In addition to the contribution of the surface structure, the chemical element in the coatings or matrix material is also one of the main factors affecting the superhydrophobic self-cleaning, antireflective and mechanical properties. The coupling of materials and structures often results in positive synergies, and it is necessary to reasonably control the composition and rough structure of the coating surface [29,30,31]. In addition, the problem of coating abrasion is unavoidable, and the application of most coatings is limited by the problem of poor frictional properties. Hence, ensuring antireflective and superhydrophobic self-cleaning properties while improving the mechanical stability of coatings has become a key issue in the field.

In order to solve the above problems and further improve the multifunctional properties of the aluminum alloy surface, matte polyurethane was used as the light-absorbing material, SiO_2_ nanoparticles (SiO_2_ NPs) as the structural material, and PDMS as the binder to prepare SiO_2_/PDMS/matte polyurethane biomimetic composite coating (BCC). This BCC possesses a composite structure similar to the morphologies of lotus leaves, exhibiting satisfactory antireflective and superhydrophobic self-cleaning properties. Moreover, the BCC also has good mechanical stability, chemical stability, and thermal stability, providing feasible solutions for the industrial application of functional treatment of aerospace and other devices and equipment surfaces.

## 2. Materials and Methods

### 2.1. Materials

SiO_2_ nanoparticles (200 nm) were prepared by the Stöber sol-gel method [32] (Appendix A). Matte polyurethane (main lacquer and curing agent), hexamethyldisilazane (HMDS) was purchased from Shanghai Fuji Industrial Co., Ltd., Shanghai, China. Polydimethylsiloxane (PDMS 184, including elastomer and curing agent) was obtained from Dow Corning Corporation Co., Ltd., Midland, MI, USA. Ethyl acetate (A.R.) and heptadecafluorodecyltrimethoxysilane (FAS-17, A.R.) were supplied by Shanghai Aladdin Biochemical Technology Co., Ltd., Shanghai, China. Sodium hydroxide (A.R.) and hydrochloric acid (A.R.) were supplied by Beijing Chemical Factory, Beijing, China. Anhydrous ethanol (A.R.) was provided by Sinopharm Chemical Reagent Co., Ltd., Shanghai, China. Deionized water was provided by Shanghai Bohu Biotechnology Co., Ltd., Shanghai, China.

### 2.2. Preparation of the BCC

#### 2.2.1. SiO_2_/PDMS/Matte Polyurethane Dispersion

As shown in Figure 1A, there is a micro-/nano-composite functional structure on the surface of the lotus leaf, which exhibits an average reflectance of less than 6.5% (450–1000 nm) and a CA of over 157°. Inspired by the excellent antireflective and self-cleaning properties of lotus leaf, the BCC based on SiO_2_/PDMS/matte polyurethane was designed, and the specific preparation process is shown in Figure 1B. First, 3.0 g PDMS (mixed with Sylgard 184 elastomer and curing agent in a mass ratio of 10:1) was added to 20 mL of ethyl acetate, and stirred magnetically for 15 min at room temperature. Then, 1.5 g modified SiO_2_ NPs (200 nm) was weighed and added to the above mixed solution, followed by stirring magnetically for 45 min at room temperature. After the SiO_2_ NPs are completely dispersed, 4.5 g matte polyurethane was added into the above mixed solution with the mass ratio of component A (main paint):component B (curing agent) = 5:1, and stirring magnetically for 60 min at room temperature. Subsequently, 4 μL of heptadecafluorodecyltrimethoxysilane (FAS-17) was added to the above mixture, followed by stirring magnetically at room temperature for 40 min. After that, this mixed dispersion was sprayed onto the surface of the aluminum alloy with laboratory spraying device.

#### 2.2.2. The Pretreatment Process of the Aluminum Alloy Surface

First, sandpapers (800 cW, 1500 cW and 2000 cW) are used to polish the 6061 aluminum alloy, and the sample with a surface roughness of <0.05 μm was obtained. Then, the sample was ultrasonically treated with absolute ethanol and deionized water for 10 min to remove surface residues of impurities, and dried with nitrogen. The diameter of the nozzle hole of the spray device is about 1 mm, and the spray distance between the nozzle and the aluminum alloy substrate is about 15 cm. The spraying process was made 20 times in 60 s at room temperature, repeated for 20 times. During the spraying process, the spray unit is moved back and forth in order to obtain a relatively uniform coating. Finally, the obtained sample was dried at 70 °C for 2 h to obtain the SiO_2_/PDMS/matte polyurethane BCC.

### 2.3. Characterization

The surface morphology of the coatings was observed by field emission scanning electron microscope (SEM, JSM-6700F, JEOL, Tokyo, Japan). To prepare the sample, the coating is sprayed directly and cured on a 1 cm × 1 cm substrate material, which is then glued to the vertical sample stage of the SEM by conductive glue, and then they were treated with gold spraying for 240 s to increase the surface conductivity and then observed in vertical view. The element composition of the coating surface was analyzed with the attached energy chromatograph (EDS, OXFORD X-MaxN 150, Oxford Instruments, Abingdon, UK). The chemical elements on the surface of the coating were analyzed by X-ray powder diffractometer (XRD, Bruker D8, Billerica, MA, USA) at a test angle of 10–80°, and the scanning speed is 4°/min. The chemical bonds of the coatings were analyzed by X-ray photoelectron spectroscopy (ULTRA DLD, Shimadzu Ltd., Kyoto, Japan). The reflectance spectrum of the coating surface was measured by a fiber optic spectrometer (Ocean Optics USB 4000, Winter Park, FL, USA). The CA and sliding angle (SA) of the coating surface were obtained by an optical contact angle meter (OCA 20, Dataphysics, Filderstadt, Germany). Five measurements were performed on each sample surface and the average value was taken. The thermal stability of the coating was measured by a Japanese Rigaku thermogravimetric analyzer (TG-DTA8122, Rigaku, Wilmington, MA, USA), the heating rate was 15 °C/min, the temperature range was 0–1000 °C, and the atmosphere was nitrogen. A high-speed camera (Phantom v711, Vision Research, Inc., Charlottetown, PE, Canada) was used to observe the bouncing process of water droplets on the surface of the BCC. All optical photos were taken with a 7D Canon camera (DS126251). The microscopic morphology of the bioinspired AR film was observed by using the field emission scanning electron microscope (SEM) of ZEISS Company, Oberkochen, Germany.

## 3. Results and Discussion

### 3.1. Morphologies and Composition Analysis of BCC

As shown in Figure 2, the morphologies of the coating are similar to lotus leaf hierarchical structures. Due to the presence of matte polyurethane, the SiO_2_ NPs exhibit a certain agglomeration phenomenon, forming the micro-/nano-structures. Since the material ratios will influence the interfacial morphology and properties of the prepared coatings, a series of BCCs with different single components specific gravity were fabricated. The corresponding antireflective and self-cleaning properties were analyzed to obtain the optimal material proportioning of 4.5 g for Matte polyurethane, 1.5 g for SiO_2_ NPs and 3.0 g for PDMS (Appendix A). The surface structure of the optimized biomimetic coatings was observed to have different structures at the micron and nano scales, showing a distinctly composite morphology (Figure 2a–c), with a surface roughness *R_a_* of 0.434 μm (Figure 2d) and a relatively homogeneous structure distribution in vertical direction (Figure 2e).

Figure 3a–c show the XPS spectrum scanning and the high-resolution scanning spectrum of the BCC modified by FAS-17. It can be seen from Figure 3a that the coating mainly contains four elements, F, O, C, and Si, and their binding energies correspond to the positions of 688.99 eV, 532.80 eV, 284.06 eV, and 103.04 eV, respectively. They correspond to peak spectra of F 1s, O 1s, C 1s, and Si 2p. A strong characteristic peak of F 1s can be clearly observed from the high-resolution scanning spectrum of XPS (Figure 3b), which is mainly due to the CF and CF_2_ bonds of FAS-17. In addition, the stronger characteristic peaks corresponding to Si 2p, and it belongs to SiO_2_ NPs, PDMS, and FAS-17 (Figure 3c). Further, it can be seen from Figure 3d that a broad diffraction peak appears at 2θ = 20.86°. Comparison with the standard card of characteristic peaks of silica (JCPDS29-0085) confirms that SiO_2_ NPs exists mainly as amorphous silica before and after modification [33]. In addition, several other diffraction peaks appeared on the curve, and these peaks were related to the mixed material of the matte polyurethane. The diffraction peak at 2θ = 22.78° was the dispersion peak of the irregular crystal region of the hard segment of the polyurethane material [34]. The above-mentioned EDS, XPS, and XRD results can confirm that Matte polyurethane, SiO_2_ NPs, and PDMS are successfully included in the BCC.

### 3.2. Antireflective and Self-Cleaning Properties of BCC

Compared to the uncoated substrate surface, the BCC reduces the average reflectivity of the substrate material from 60% to 10% (7.64% minimum reflectivity) in 450–1000 nm wavelength band, displaying significant broadband antireflection characteristics (Figure 4a). Meanwhile, the static CA of the surface increases from 139.3 ± 0.75° (without SiO_2_) to 156.32 ± 0.85° (with SiO_2_), demonstrating that the structure produced by SiO_2_ has a significant hydrophobic promoting effect. The dynamic water SA was then measured to be only 4.9 ± 0.65°, indicating that water droplets can be rolled off the coated surface quickly.

Figure 5a,b show the self-cleaning performance test of SiO_2_/PDMS/matte polyurethane BCC. When the test sample is placed on the glass sheet with an angle of less than 10°, the water droplets on the surface of the coating completely roll off the surface in the form of a ball, leaving no residue of water droplets or pollutants. It indicates that the coating has excellent self-cleaning ability and low adhesion, which is very similar to the self-cleaning performance of lotus leaves.

In order to further verify the excellent superhydrophobic property of the BCC, a dynamic test of the droplet bounce on the BCC surface was recorded with high-speed photography (Figure 5c). During the test, 1.40 mm radius water droplets were set to fall from a height of 30 mm. When water droplets collide with the coating interface, they shrink and completely bounce off the coating surface at approximately *t* = 17.2 ms without any residue. It indicates that the short acting time of water droplets on the coating interface provides perfect evidence for the coating to achieve good waterproof properties. In order to further verify this view, the theoretical contact time of water droplets hitting the surface is calculated with Equation (1) [35]:(1)$$t_{c}=2.6\sqrt{\rho D_{0}^{3}/\left(8\gamma \right)}$$
where ρ is the density of the drop, *D*_0_ is the drop diameter and *γ* is the surface tension of the drop. Since the *γ* = 71.97 × 10^−3^ N·m^−1^ [35], the theoretical contact time between the droplet and the coating surface was calculated to be *t_c_* = 16.05 ms. As mentioned, actual contact time is 17.2 ms, and these two contact times were quite closed. It indicates that SiO_2_/PDMS/matte polyurethane BCC possessed excellent superhydrophobic property.

### 3.3. Antireflective and Self-Cleaning Mechanism

The structure of the surface roughness and the low surface energy of the material applied to the surface are critical to the wetting performance of a coating (Figure 6A). The BCC prepared in this experiment can achieve superhydrophobic self-cleaning characteristics mainly due to the combination of SiO_2_ NPs, PDMS and matt polyurethane to generate irregular micro/nano disordered rough structure through chemical bond, and the hydrophobic fluorocarbon chain on FAS17 modifier is successfully bonded to the surface of BCC.

The modification of BCC with a micro/nano-structure using low surface energy materials allows a large amount of air to be trapped in the grooves or cavities of the surface. In this case, the surface wettability can be described by Cassie-Baxter model in Equation (2) [36]:(2)$$\cos\theta_{c}=f\cos\theta +\left(1-f\right)\cos180^{{}^{\circ}}=f\cos\theta +f-1\;$$
where *θ_c_* is the apparent contact angle of structural surface, while *θ* is the contact angle of the smooth surface. *f* denotes the fraction of solid surface wetted by the liquid at the liquid-solid interface. The area fraction at the liquid-gas interface can be expressed as 1 − *f*.

The CA of the smooth matte polyurethane/PDMS mixed coating surface is *θ* = 139.3 ± 0.75° (Figure 4), and the CA of the BCC surface is *θ_c_* = 156.32 ± 0.58° (Figure 4). The contact area between the droplets and the coating surface can be calculated as *f* = 0.347. This means that approximately 34.7% of the droplet area is in contact with the coating surface and approximately 65.3% of the area is completely in contact with the air. It indicates that more air is trapped between the micro/nano-structures, reducing the contact area between the droplets and the surface, and allowing the coating to achieve remarkable superhydrophobic performance. HMDS was used for surface fu6ctionalization, and the surface groups of SiO2 NPs were replaced with methyl groups. On the one hand, it prevents NPs from agglomerating seriously during mechanical stirring, and more importantly, it endows the surface a hydrophobic effect. On the other hand, after modification by FAS-17, a large number of hydroxyl groups originally present on the surface are replaced by fluorocarbon chains, which greatly improves the surface wetting characteristics.

For optical properties, when incident light acts on the surface of the BCC, it first contacts with the agglomerated SiO_2_ micro/nano structure, resulting in refraction, reflection, transmission and scattering (Figure 6B). Most of the light propagates inside the BCC, and finally is absorbed by BCC or transmitted to the aluminum alloy substrate under the joint action of various optical mechanisms. However, excessive surface roughness may increase the obvious scattering effect, reducing the light absorption of the coating. The optimized BCC achieves excellent antireflective performance, which is mainly due to the joint action of matte polyurethane and internal micro-/nano-structure in the coating. Although matte polyurethane itself has good light absorption performance, the coating without SiO_2_ NPs only achieves a reflectivity of about 20.91%, while the surface reflectivity decreased to about 13.47% after adding SiO_2_ NPs (Appendix A). It can be seen that the presence of this lotus-like surface hierarchy plays a crucial role in improving the anti-reflective and superhydrophobic properties of the coating surface.

### 3.4. Mechanical Durability

Figure 7a shows the process of abrasion test for the BCC surface on sandpaper. First, the bottom surface of the aluminium alloy coated with BCC was attached to one side of the slide. Subsequently, the upper surface of the coating was placed on sandpaper (1500 cW) so that the coating surface was in full contact with the sandpaper. Then, a 20 g weight was placed on the upper surface and the specimen was dragged in the direction of the yellow arrow with a rate of 3–5 mm/s for a distance of 10 cm for each abrasion. As shown in Figure 7b, the surface CA of the coated surface increased to 160.68 ± 0.77° after four abrasion tests. When the abrasion tests reached 44 times, the CA still managed to reach 161.02 ± 1.73°. Although the SA of the BCC shows a gradual increase trend with the number of abrasions, it still maintains satisfactory self-cleaning property with a SA of 8.7 ± 1.20° after 44 abrasion cycles. The reason for this phenomenon is that the coating surface tends to a relatively flat state due to the surface tension of the uncured coating material. After the abrasion test, on the one hand, excess PDMS can be removed, resulting in roughness of the coating increasing. The air trapped between the structures also increases, which can reduce the contact area between the water droplets and the coating surface. On the other hand, the abrasion surface exposes hydrophobic SiO_2_ NPs, exposing more hydrophobic groups and thus increasing hydrophobicity. The overall factors result in an increase in the CA and the decrease in the SA. The results show that the BCC has good wear resistance. In Figure 7c, the average reflectivity of the BCC after 44 times abrasion reaches around 20% in the wavelength range of 480–1000 nm, which still displays obvious antireflective performance. In contrast, the reflectivity of severely worn areas (destroyed point) reaches around 35%. In general, the coating maintains good antireflective properties even within 44 times abrasion cycles. When a small loss of composite structure occurs, the surface self-grows in situ to a morphology similar to the original structure due to the presence of micro-/nano-particles, so that there is no substantial increase in reflectance. When the coating wears excessively, the loss of a large amount of composite material from the surface will result in a reduction of light absorbing material in the coating, meanwhile reducing the coating thickness of the micro/nanostructure and shortening the action of the incident light between the structures. As a result, the incident absorption of the coating was reduced. Figure 7d shows the surface morphology of the coating after abrasion. After 44 abrasions, the area of the broken area is not very large and the remaining coating material is still well attached to the substrate surface by the action of the adhesive and the overall abrasion resistance is excellent.

Figure 8a evaluates the surface wettability of the BCC before and after the tape stripping test, and it shows that the surface still exhibits good superhydrophobic characteristics after 230 tape stripping tests. This indicates that only local areas of the coating will show slight surface flaking (marked by red dashed lines in Figure 8b), and new micro-nano structures will be formed in-situ in this area. Similarly, the surface retained good superhydrophobic properties when the BCC was subjected to 210 scraping tests (Figure 8c). In particular, only a few areas on the surface of the coating after scraping showed the phenomenon of coating peeling, while most of the coating material remained in good contact with the substrate (Figure 8d). These results show that the coating possesses excellent adhesion resistance and mechanical robustness. In addition, the self-cleaning properties of the knife scraper sample were tested. It was found that when water drops fell onto the white CaCO_3_ powder (placed in the test area), the drops still rolled off in a spherical pattern and carried away the powder with no residual water drops or white powder left on the surface of the coating. This indicates that the coating still maintains good superhydrophobic self-cleaning properties after the test (Figure 8e,f).

### 3.5. Stability Test of BCC

Figure 9 shows the variation curve of the CA of the BCC with different pH values. The volume of the droplet in the test is 8 μL. When pH = 1, the CA of the BCC is 148.18 ± 0.94°. With the increase of pH value, the surface CA is more than 150°, and the CA of BCC is 150.64 ± 0.46°when pH = 7. However, the CA varies from 149° to 150° with the increase of pH value. When pH = 14, the surface CA becomes 147.94 ± 1.26°. It can be seen that the coating displays obvious rejection of acidic and alkaline solutions. Although it exhibits relatively poor rejection of strong acids and bases (pH = 1 and pH = 14), the BCC still shows a hydrophobic state.

Figure 10a shows the variation curve of surface CA and SA after the coating is heated in a vacuum drying oven for 2 h (test temperature: 70–150 °C). With the increase of temperature, the CA of coating surface varies from 157.57 ± 0.65° to 159.67 ± 2.48°, while the SA varies from 4.9 ± 1.52° to 6.3 ± 1.04°, which indicates that the temperature change will not have an obvious effect on the wettability of the coating surface.

In order to further investigate the thermal stability characteristics of the coating, thermogravimetric test analysis was carried out (Figure 10b). The weight ratio curve of the coating decreases slightly between 0 °C and 269.7 °C, which is mainly due to the evaporation of the absorbed water on the surface of the coating. The curve decreases from 269.7 °C to 537.6 °C may be due to the decomposition of C-O bond of urethane group in matte polyurethane, which forms isocyanates and polyols, and then further decomposed into amines. In this temperature range, the -OSi(CH_3_)_3_ groups on the SiO_2_ surface and the -CH_3_ groups on the PDMS also begin to be decomposed, resulting in a loss of curve mass. Then, the weight drops sharply from 537.6 °C to 661.4 °C, which is caused by the decomposition of Si-C bonds in the coating and the decomposition of other components in the matte polyurethane. The temperature curve gradually becomes stable from 661.4 °C to 1000 °C due to the residue of the remaining components in the matte polyurethane and the existence of Si-O-Si structural bonds in SiO_2_, and the residual weight ratio is about 36.23%. These results show that the BCC is able to withstand the temperatures up to 200 °C, providing a favorable guarantee for the engineering application of functional coatings on aluminum alloy materials in most application scenarios.

## 4. Conclusions

In this study, we successfully developed a hierarchically structured BCC with antireflection and superhydrophobic self-cleaning characteristics on an aluminum alloy surface through a one-step spray-deposition process. The advantage of this biomimetic construction strategy enabled SiO_2_ NPs to be firmly encapsulated in the mixture of PDMS and matte polyurethane, endowing the BCC with robust functional characteristics. The effects of the amounts of SiO_2_ NPs, PDMS, and matte polyurethane on the antireflection and wettability properties of BCC were discussed separately. Meanwhile, the optimal ratio of SiO_2_ NPs, PDMS, and matte polyurethane to form BCC was determined to be 1.5 g, 3 g, and 4.5 g, respectively. It also showed a CA of 156.32 ± 0.58°, SA of 4.9 ± 0.65° and minimum reflectivity of approximately 7.64% in the wavelength range of 450–1000 nm. Compared with uncoated aluminum alloy surface, the obtained BCC effectively achieved the synergistic construction of antireflection and superhydrophobic self-cleaning properties on the substrate surface. In addition, thanks to the synergistic action of SiO_2_ NPs, PDMS and matte polyurethane, the BCC exhibited excellent mechanical stability, good acid resistance. In particular, the BCC could withstand 44 abrasion tests, 230 tape stripping tests, and 210 scraping tests. Importantly, after testing, the BCC still showed good antireflection and superhydrophobic self-cleaning properties. This study provides an effective reference solution for solving the common problems faced by the engineering and industrial applications of functional coatings.

## Figures and Tables

**Figure 1 nanomaterials-13-01855-f001:**
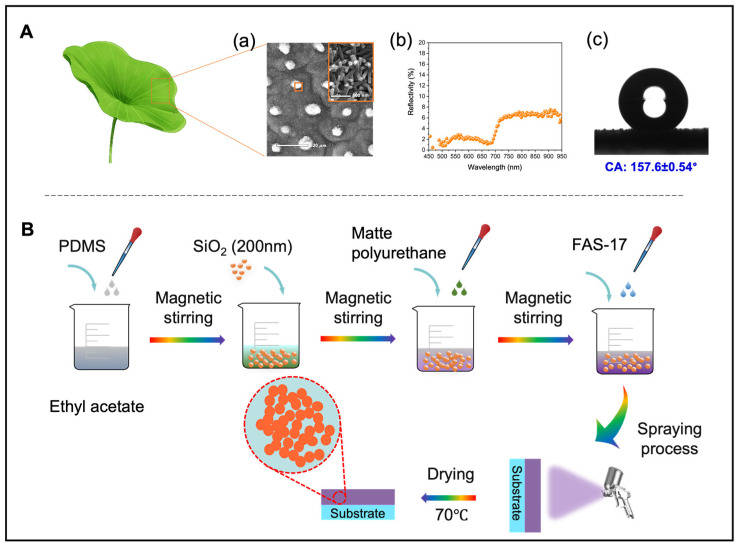
(**A**) Excellent anti-reflection and self-cleaning properties of lotus leaf composite structures: (**a**) SEM image, (**b**) Reflectivity, (**c**) Static CA of the micro-/nano-composite structure. (**B**) Schematic diagram of the preparation process of SiO_2_/PDMS/matte polyurethane BCC.

**Figure 2 nanomaterials-13-01855-f002:**
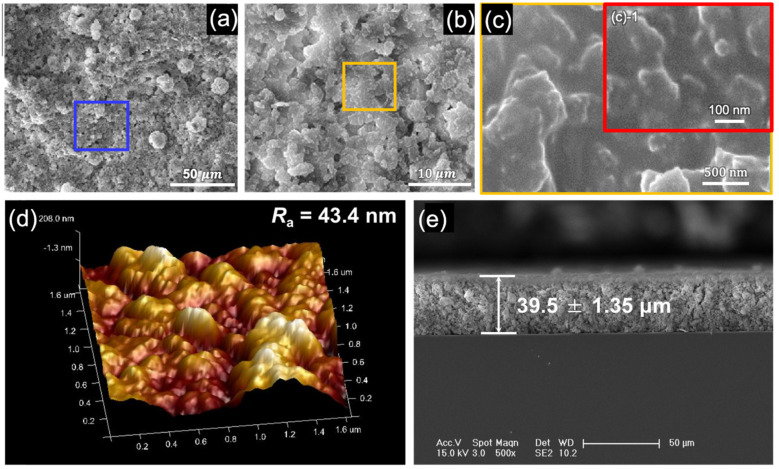
(**a**–**c**) SEM images of the micro-/nano-structures of the BCC with scale bars of 50 μm, 10 μm, 500 nm, respectively. (c)-1 is local area enlargement of figure (**c**). (**d**) AFM image of the coating. (**e**) SEM image of the cross-section of the coating surface.

**Figure 3 nanomaterials-13-01855-f003:**
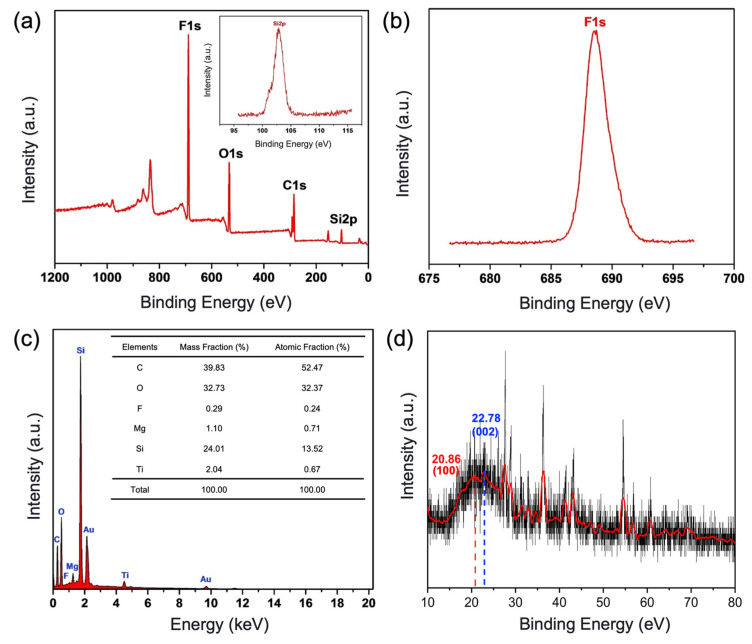
Chemical composition analysis of the BCC. (**a**) XPS full spectral scan of the coating and high-resolution scanning spectra of the S 2p of the coating. (**b**) High-resolution scanning spectra of the F 1s of the coating. (**c**) EDS spectra of the coating. (**d**) XRD map of the coating.

**Figure 4 nanomaterials-13-01855-f004:**
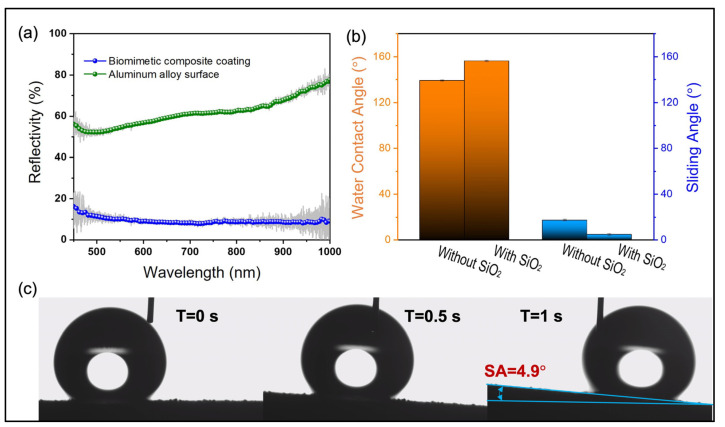
Optical and wettability characterization. (**a**) Reflectivity of substrates with/without biomimetic functional coating. (**b**) CA of the coating surface with/without SiO_2_. (**c**) Water droplet roll-off process on BCC.

**Figure 5 nanomaterials-13-01855-f005:**
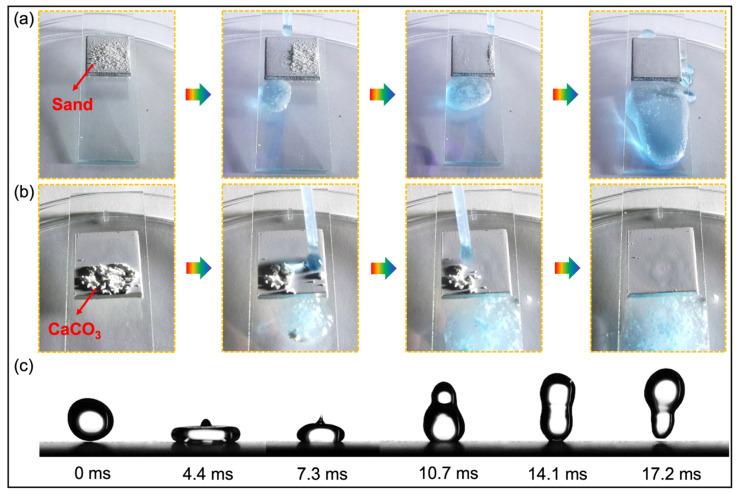
Surface self-cleaning performance and droplet bounce test of BCC. (**a**) Fine sand and (**b**) white calcium carbonate powder was used as pollutants for self-cleaning tests. (**c**) Bounce dynamics test of water droplet hitting coating surface.

**Figure 6 nanomaterials-13-01855-f006:**
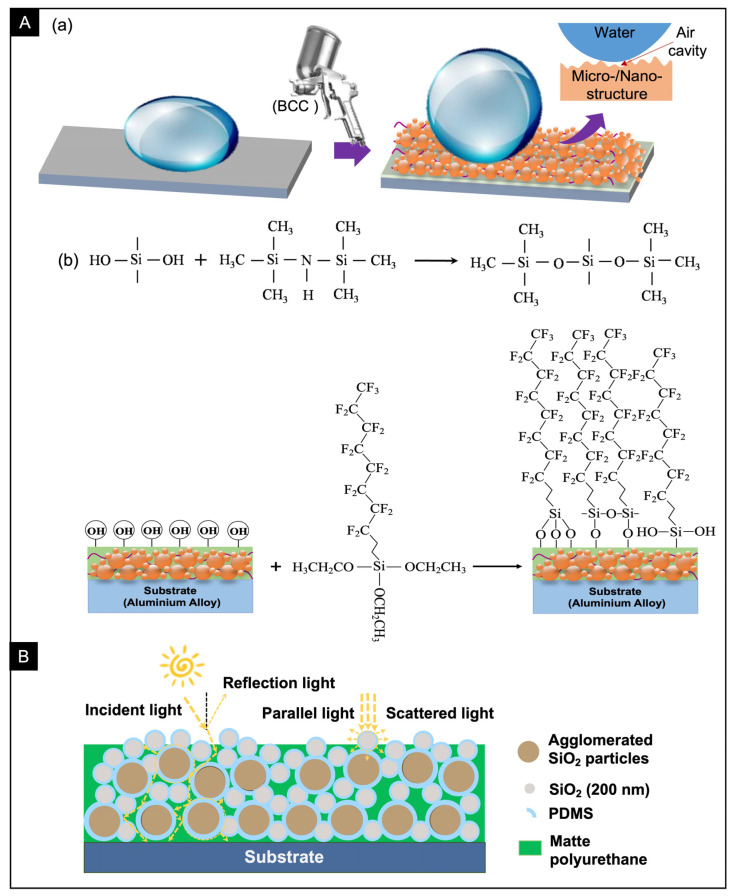
(**A**) (**a**) Schematic diagram of the mechanism of the BCC surface to realize superhydrophobic self-cleaning and anti-reflection characteristics. (**b**) Antireflective mechanism of the coating surface. (**B**) Anti-reflection mechanism of bionic coating.

**Figure 7 nanomaterials-13-01855-f007:**
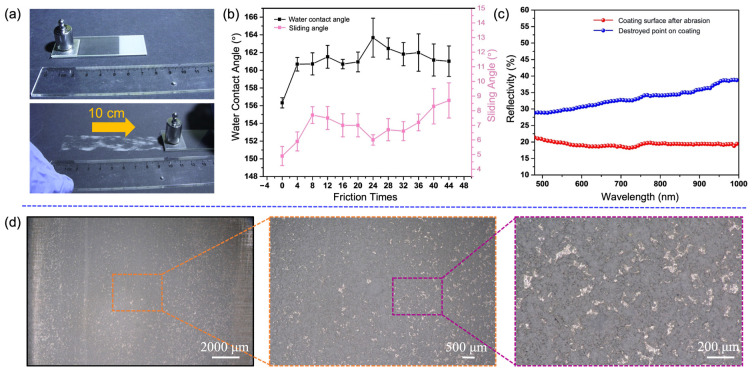
Frictional abrasion resistance test tests on the surface of the BCC. (**a**) Frictional abrasion test procedure. (**b**) CAs and SAs. (**c**) Reflectivity of the coated surface after abrasion. (**d**) Morphologies of BCC after 44 times abrasion.

**Figure 8 nanomaterials-13-01855-f008:**
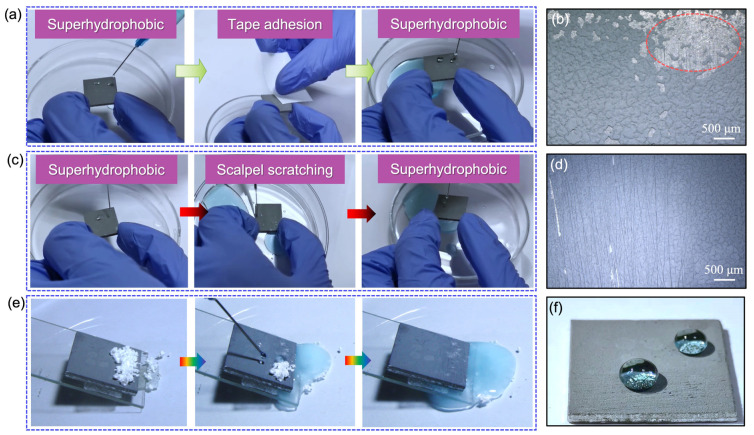
Tape stripping test and knife scraping test of BCC. (**a**) Tape stripping. (**b**) Ultra-depth-of-field microscopic images of the coating surface morphology after tape stripping test. (**c**) Knife scraping tests. (**d**) Ultra-depth-of-field microscopic images of the coating surface morphology knife scraping test. (**e**) Self-cleaning test of coating surface after tape stripping test and knife scraping test. (**f**) Superhydrophobic characteristics of the sample surface after testing.

**Figure 9 nanomaterials-13-01855-f009:**
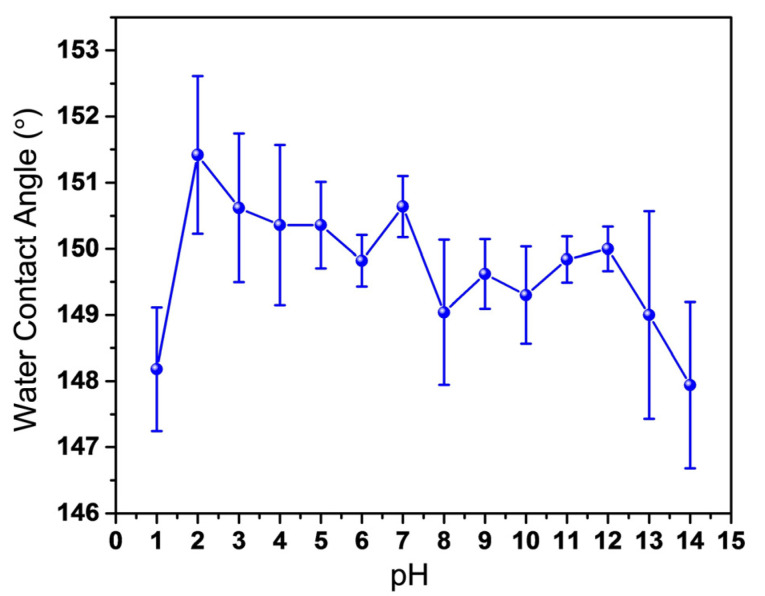
CA of BCC with different pH values.

**Figure 10 nanomaterials-13-01855-f010:**
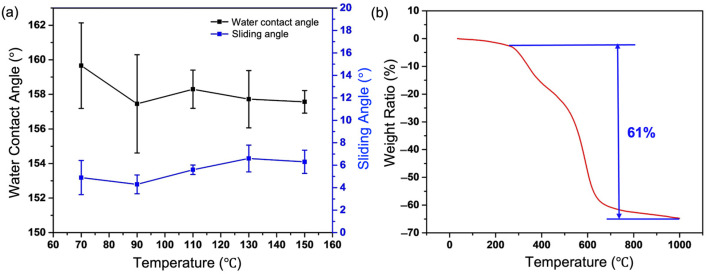
BCC surface resistance to high temperature and thermogravimetric test. (**a**) The variation curve of coating surface with different temperature values. (**b**) Thermogravimetric analysis curve of coating.

## Data Availability

Not applicable.

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
