# Peer review of "Multifunctional Biomimetic Composite Coating with Antireflection, Self-Cleaning and Mechanical Stability"

_nanomaterials, 2023, doi:10.3390/nano13121855_

Round 1

Reviewer 1 Report

Appreciate the authors for contributing an academic paper in the field of superhydrophobic surfaces.

A few minor comments and Suggestions as follow:

1. Page 5, It needs to describe of what 'JCPDS file' is.

2. Page 9, After the friction test, the contact angle increased, but the explanation for the reason seems to be needed in the text. Please also explain why CA and SA change.

The manuscript is nicely, concisely and adequately written. Apart from small spelling mistakes, which will be fixed in the proofing stage, I would recommend that the figures should be larger, or at least the text.

Author Response

Point-by-Point Response:

Reviewer 1

Comments: Appreciate the authors for contributing an academic paper in the field of superhydrophobic surfaces.

A few minor comments and Suggestions as follow:

  1. Page 5, It needs to describe of what 'JCPDS file' is.

Response:

Thank you for your comments. JCPDS stands for joint committee on powder diffraction standards, which is a standard database where the diffraction spectra obtained from XRD tests can be compared with it to characterize the structural phases of materials. The full name of the 'JCPDS file' has been added in the revised manuscript on page 5 line 177-178.

  1. Page 9, After the friction test, the contact angle increased, but the explanation for the reason seems to be needed in the text. Please also explain why CA and SA change.

Response:

Due to the surface tension of the uncured material, coating surface tends to a relatively flat state. After the friction test, on the one hand, excess PDMS can be removed, resulting in roughness of the coating increasing. And the air trapped between the structures also increases, which can reduce the contact area between the water droplets and the coating surface. On the other hand, the friction surface exposes hydrophobic SiO2 NPs, exposing more hydrophobic groups and thus increasing hydrophobicity. The overall factors result in an increase in the CA and the decrease in the SA. It has been explained in the revised manuscript on page 10 line 283-290.

Reviewer 2 Report

The research presented in this article is interesting, with some pertinent results. It is written in an understandable form. The macro objective is to obtain a high-performance antireflection and self-cleaning surface improving the mechanical stability. In my opinion, the article can be accepted for publication after answering the following doubts:

In lines 111-112 it is not clear if the spraying process was made 20 times in 60 seconds or each spray is made continuously for 60 seconds and then repeated 20 times. If this last hypothesis was used, which was the time and temperature of the substrate between depositions (spray).

In lines 120 and further, the authors must add the description of the sample preparation for the cross section analysis by SEM microscopy.

Figure 3d should have a better resolution. Is it possible to repeat the measure increasing the accumulation time ?

In lines 327-342: the Tg results show a high weight loss with temperature, which should be related with irreversible structural changes. It is not clear how it is concluded that the sample can stand for “high-temperatures”. This should be reanalysed. It will be excellent to add new results on samples that were subjected to a heating process above 700 ºC (the thermal zone where it is affirmed that the sample is stable).

The abrasion tests, the tape stripping tests and the scrapping test were made under some international standard? If not, the 44 abrasion tests, the 230 tape stripping tests, and the 210 scraping tests were defined in accordance with which condition? 

Author Response

Point-by-Point Response:

Reviewer 2

Comments: The research presented in this article is interesting, with some pertinent results. It is written in an understandable form. The macro-objective is to obtain a high-performance antireflection and self-cleaning surface improving the mechanical stability. In my opinion, the article can be accepted for publication after answering the following doubts:

  1. In lines 111-112 it is not clear if the spraying process was made 20 times in 60 seconds or each spray is made continuously for 60 seconds and then repeated 20 times. If this last hypothesis was used, which was the time and temperature of the substrate between depositions (spray).

Response:

Thank you for your guidance on our experimental formulation. Here, the spraying process was made 20 times in 60 s at room temperature. These experimental details have been added in page 3 line 120-121.

  1. In lines 120 and further, the authors must add the description of the sample preparation for the cross-section analysis by SEM microscopy.

Response:

The coating is sprayed directly and cured on a 1 cm x 1 cm substrate material, which is then glued to the vertical sample stage of the SEM by conductive glue, and then they were treated with gold spraying for 240 s to increase the surface conductivity and then observed in vertical view. This description of the sample preparation has been added in page 4 line 130-134.

  1. Figure 3d should have a better resolution. Is it possible to repeat the measure increasing the accumulation time? 

Response:

Thanks for your patient suggestion. We have replaced the higher resolution image in the revised manuscript on page 6, which used the normal scanning speed of 4°/min. And it also meets the experiment requirements.

  1. In lines 327-342: the Tg results show a high weight loss with temperature, which should be related with irreversible structural changes. It is not clear how it is concluded that the sample can stand for “high-temperatures”. This should be reanalysed. It will be excellent to add new results on samples that were subjected to a heating process above 700 ºC (the thermal zone where it is affirmed that the sample is stable).

 Response:

Thank you for your crucial comment. The issue you point out is very professional. The “high temperature resistance (up to 200 ℃)” we mentioned is only relative to the temperature of conventional environmental conditions, and it is not strictly high-temperature resistance. Therefore, the original expression "high-temperature" is inappropriate, now we have revised this statement as: ” These results show that the BCC is able to withstand the temperatures up to 200 ℃, providing a favorable guarantee for the engineering application of functional coatings on aluminium alloy materials in most application scenarios.” in page 12 line 362-365.

  1. The abrasion tests, the tape stripping tests and the scrapping test were made under some international standard? If not, the 44 abrasion tests, the 230 tape stripping tests, and the 210 scraping tests were defined in accordance with which condition? 

Response:

Thank you for your very professional question. The abrasion tests, the tape stripping tests and the scrapping test in this research are qualitative, and they refer to the research of some others[1-4]. Strictly speaking, there is no common international standard about these tests. As for the abrasion tests (sandpaper: 1500 cW, weight: 20g), the specimen was dragged in same direction with a rate of 3-5 mm/s for a distance of 10 cm for 44 times abrasion. As for the tape stripping tests, using ordinary double-sided tape, after manual pasting and receiving 230 times, the sample used for qualitative analysis still has good performance (CA, reflectivity). As for the scraping tests, a scalpel was used to randomly scratch the surface of the coating for 210 times, after which the sample performance was tested with little change. These experiments are qualitatively analyzed, providing some references for readers.

References:

  1. Zhang, J.; Zhang, L.; Gong, X. Large-Scale Spraying Fabrication of Robust Fluorine-Free Superhydrophobic Coatings Based on Dual-Sized Silica Particles for Effective Antipollution and Strong Buoyancy. Langmuir2021, 37, 6042–6051, doi:10.1021/acs.langmuir.1c00706.
  2. Xiao, P.; Yang, L.; Liu, J.; Zhang, X.; Chen, D. A Non-Fluorinated Superhydrophobic Composite Coating with Excellent Anticorrosion and Wear-Resistant Performance. Front. Chem.2022, 10, 952919, doi:10.3389/fchem.2022.952919.
  3. Mirmohammadi, S.M.; Hoshian, S.; Jokinen, V.P.; Franssila, S. Fabrication of Elastic, Conductive, Wear-Resistant Superhydrophobic Composite Material. Sci Rep2021, 11, 12646, doi:10.1038/s41598-021-92231-x.
  4. Tang, Y.; Wu, F.; Fang, L.; Ruan, H.; Hu, J.; Zeng, X.; Zhang, S.; Luo, H.; Zhou, M. Effect of Deposition Sequence of MgAl-LDH and SiO2@PDMS Layers on the Corrosion Resistance of Robust Superhydrophobic/Self-Healing Multifunctional Coatings on Magnesium Alloy. Progress in Organic Coatings2023, 174, 107299, doi:10.1016/j.porgcoat.2022.107299.

Reviewer 3 Report

The paper "Multifunctional Biomimetic Composite Coating with Antireflection, Self-Cleaning and Mechanical Stability" has very important content and is suitable for publication in Nanomaterials after some minor corrections:

  1. The introduction has the necessary aspects regarding functional coatings, but some alternative coating methods should be added in order to make a comparison. Also, in the presentation, the influence of chemical elements in the coatings on the mentioned properties should be mentioned. Suggested reference: a) Effects of the pre-treatment with atmospheric-air plasma followed by conventional finishing, Revista de Chimie, Volume 65, Issue 6, Pages 676–683, June 2014; b) 10.3390/mi12121447;
  2. In 2.3. Characterization, please add more specifications for the SEM and XRD parameters.
  3. The legend of figures 2 (a–c) is not quite clear.
  4. Add the phase diffractogram if you mentioned that XRD analysis has been done, and comment on the phase components.
  5. The rest is fine.

Author Response

Point-by-Point Response:

Reviewer 3

Comments: The paper "Multifunctional Biomimetic Composite Coating with Antireflection, Self-Cleaning and Mechanical Stability" has very important content and is suitable for publication in Nanomaterials after some minor corrections:

  1. The introduction has the necessary aspects regarding functional coatings, but some alternative coating methods should be added in order to make a comparison. Also, in the presentation, the influence of chemical elements in the coatings on the mentioned properties should be mentioned. Suggested reference: a) Effects of the pre-treatment with atmospheric-air plasma followed by conventional finishing, Revista de Chimie, Volume 65, Issue 6, Pages 676–683, June 2014; b) 10.3390/mi12121447;

Response:

We appreciate your constructive comments. Some alternative coating methods have been added as “Ghosh et al. prepared vertically arranged tree-like nanostructures with graded roughness on polished silicon substrates, exhibiting excellent superhydrophobicity and low adhesion [3]. Wang et al. prepared ZnS superhydrophobic coating on the surface of zinc substrate by economic and environmental-friendly solvothermal method and chemical modification method [4]. Xia et al. provided a SiO2-TiO2-PDMS composite coating to demonstrates the potential application of coatings to protect architectures from detrimental atmospheric effects via a self-cleaning approach [5].” Besides, the influence of chemical elements in the coatings was emphasized on page 1 line 37-44, and the reference you mentioned also have been added.

  1. In 2.3. Characterization, please add more specifications for the SEM and XRD parameters.

Response:

As mentioned, more specifications for the SEM and XRD parameters has been added, the manuscript was revised as “The surface morphology of the coatings was observed by field emission scanning electron microscope (SEM, JSM-6700F, JEOL). To prepare the sample, the coating is sprayed directly and cured on a 1 cm x 1 cm substrate material, which is then glued to the vertical sample stage of the SEM by conductive glue, and then they were treated with gold spraying for 240 s to increase the surface conductivity and then observed in vertical view.” in page 4 line 130-134, “The chemical elements on the surface of the coating were analyzed by X-ray powder diffractometer (XRD, Bruker D8) at a test angle of 10-80°, and the scanning speed is 4” in page 4 line 136-138.

  1. The legend of figures 2 (a–c) is not quite clear.

Response:

The legends in Figure 2 have been revised as follow, and the original figure in manuscript has also been replaced.

  1. Add the phase diffractogram if you mentioned that XRD analysis has been done, and comment on the phase components.

Response:

Since the materials used in the composite coating are known, the XRD analysis here is only for verification purpose. Relevant expressions have been revised as:” Further, it can be seen from Figure 3d that a broad diffraction peak appears at 2θ=20.86°. Comparison with the standard card of characteristic peaks of silica (JCPDS29- 0085) confirms that SiO2 NPs exists mainly as amorphous silica before and after modification” on page 5 line 177-178, and more information was added to the Figure 3, and it was modified as follow.

Figure 3 XRD map of the coating.

  1. The rest is fine.

Response:

Thank you for your guidance, which helps us further improve the quality of the manuscript.
